# Vitamin A Deficiency and Its Association with Visceral Adiposity in Women

**DOI:** 10.3390/biomedicines11030991

**Published:** 2023-03-22

**Authors:** Érica Góes, Adryana Cordeiro, Claudia Bento, Andrea Ramalho

**Affiliations:** 1Faculty of Medicine, Federal University of Rio de Janeiro (UFRJ), Rio de Janeiro 21941-598, Brazil; 2Micronutrient Research Center (NPqM) of the Josué de Castro Institute of Nutrition (INJC), Federal University of Rio de Janeiro (UFRJ), Rio de Janeiro 21941-598, Brazil; 3Department of Nutrition and Dietetics, Josué de Castro Institute of Nutrition (INJC), Federal University of Rio de Janeiro (UFRJ), Rio de Janeiro 21941-598, Brazil; 4Department of Applied Social Nutrition, Federal University of Rio de Janeiro (UFRJ), Rio de Janeiro 21941-598, Brazil

**Keywords:** vitamin A, vitamin A deficiency, visceral adiposity, overweight, obesity

## Abstract

Body adiposity is associated with increased metabolic risk, and evidence indicates that vitamin A is important in regulating body fat. The aim of this study was to evaluate serum concentrations of vitamin A and its association with body adiposity in women with the recommended intake of vitamin A. A cross-sectional study was designed with 200 women divided into four groups according to Body Mass Index (BMI): normal weight (NW), overweight (OW), class I obesity (OI), and class 2 obesity (OII). The cut-off points to assess inadequate participants were retinol < 1.05 µmol/L and β-carotene < 40 µg/dL. Body adiposity was assessed through different parameters and indexes, including waist circumference (WC), waist-to-height ratio (WHtR), hypertriglyceridemic waist (HW), lipid accumulation product (LAP), Visceral Adiposity Index (VAI), and Body Adiposity Index (BAI). It was observed that 55.5% of women had low serum concentrations of β-carotene (34.9 ± 13.8 µmol/L, *p* < 0.001) and 43.5% had low concentrations of retinol (0.71 ± 0.3 µmol/L, *p* < 0.001). Women classified as OI and OII had lower mean values of β-carotene (OI—35.9 ± 4.3 µg/dL: OII—32.0 ± 0.9 µg/dL [*p* < 0.001]). IAV showed significant negative correlation with retinol (r = −0.73, *p* < 0.001). Vitamin A deficiency is associated with excess body adiposity in women with the recommended intake of vitamin. Greater body adiposity, especially visceral, was correlated with reduced serum concentrations of vitamin A.

## 1. Introduction

Obesity rates have grown progressively over the years. The most recent data indicate that in, the period from 1975 to 2016, the prevalence of obesity tripled, meaning that 650 million individuals now have obesity [1]. If current trends continue, by 2030 more than half of the adult population will be overweight [2]. It has been noted that women are gaining more prominence for having the highest rates of overweight and obesity across all age groups, when compared to men [3]. As such, obesity affects 15% of women and 11% of men worldwide [1].

This profile can be found in many countries. In the United States, women had a higher percentage of obesity (40%) compared to men (38%) [4]. An analysis of 105 different countries showed that women continued to stand out with the higher prevalence of being overweight and suffering from obesity when compared to men [5]. Among men, the rate is lower, reaching 21.8% and 57.5%, respectively [6]. It is estimated that if there is a reduction of at least 5% in the Body Mass Index (BMI) at a population level, in 2030 there would be a reduction in medical costs related to obesity of at least EUR 495 million over the next 20 years [7].

Following global trends, in Brazil, for 17 years (2002–2019), women have had the highest rates of obesity and being overweight. The most current national statistics shows that 29.5% of women suffer from obesity (approximately one in every three women) and 62.6% are overweight [6].

Excess body adiposity is an important risk factor for the development of Chronic Non-Communicable Diseases (NCDs), including Type 2 Diabetes Mellitus (DM2), some types of cancer, and cardiovascular disease (CVD) [1]. Body adiposity is commonly classified using the BMI [1]. However, it is known that BMI has limitations in measuring body adiposity and identifying its distribution and that such distribution is strongly correlated with the development of comorbidities and metabolic risk [8].

Besides BMI, other parameters for assessing body adiposity support the prognosis of metabolic risks resulting from excess fat mass, especially visceral fat mass, and its associated diseases.

Evidence points to vitamin A (VA) as an important regulator of body fat reserve through actions on nuclear receptors in both liver and adipose tissue [9]. Retinol metabolites regulate body adiposity since most of their effects are anti-adipogenic; they inhibit adipocyte differentiation and intracellular lipid accumulation [10].

Findings in the literature show that higher BMI is related to lower serum concentrations of VA, suggesting that vitamin A deficiency (VAD) may be related to excess body weight [11,12]. Some other studies have shown that lower VA intake was positively associated with excess adiposity [13,14]. There are also studies showing lower serum concentrations of VA in individuals with obesity compared to normal weight individuals. They also suggest that inadequate dietary intake of this vitamin is the primary cause of VAD, but no assessment of the dietary intake of this nutrient has been performed [15].

Due to the scarcity of studies evaluating the relationship between VA and body adiposity, especially visceral adiposity, the current study aims to assess serum concentrations of VA and its relationship with body adiposity by using Waist Circumference (WC), Waist-to-Height Ratio (WHtR), Hypertriglyceridemic Waist (HW), Body Adiposity Index (BAI), Visceral Adiposity Index (VAI), and Lipid Accumulation Product (LAP) as complementary parameters to assess body adiposity in women with the recommended intake of this vitamin.

## 2. Materials and Methods

This is a descriptive cross-sectional study conducted from January 2019 to October 2021, comprising adult women in different BMI classes who attended the Nutrition Service of a Municipal Health Unit, located in Rio de Janeiro.

To participate in the study, women were required to be aged <20; 59>. Exclusion criteria were unmet dietary recommendation for VA, pregnant and lactating women, those with liver diseases (except non-alcoholic fatty liver disease), disabsorptive syndromes and surgeries, acute infection, excessive alcohol intake (≥45 g), chemical dependence, nephropathies, acquired immunodeficiency syndrome, cancer, and/or the use of supplements containing VA in the six months prior to collection. This study was approved by the Research Ethics Committee of Hospital Universitário Clementino Fraga Filho, Federal University of Rio de Janeiro, Brazil (Research Protocol number 011/15-CEP).

### 2.1. Sample Size

The sample size was calculated based on a national study that assessed the Brazilian prevalence of micronutrient inadequacy (POF 2017–2018) [16]. Based on these findings, the prevalence of inadequacy for vitamin A in the female population is 80.1%.

To obtain the sample size with a 95% confidence interval, considering the prevalence of adequacy of 20% and with a sampling error of 5%, 148 women with the recommended daily food intake according to the Institute of Medicine [17] would be needed to conduct the present study.

### 2.2. Selection of Study Participants

The first phase of the research consisted of the application of the exclusion criteria; it was carried out to assess the level of vitamin A consumption. For those who met the established profile, there was clarification about the objectives and procedures of the study; then, the Informed Consent Form was provided to the women who agreed to participate in the study, who then signed and returned the form.

After applying the exclusion criteria (Figure 1), 849 women who did not meet the required criteria exited the study, and those who met the recommendation for daily VA intake continued (*n* = 231).

#### Assessment of Vitamin A Dietary

The assessment of vitamin A dietary intake was carried out using the three-day food intake record (two during the week and one on the weekend); this record quantified the average of vitamin A dietary. Additionally, the study used the 24 h recall and the food consumption frequency questionnaires.

The 24 h recall was used to obtain information about the consumption of vitamin A on the day before the interview. It considered the preparation of food and information on the weight and size of portions in grams, milliliters, and in household measurements. During data collection, each woman was instructed to ensure that no meals or snacks were missed. The primary purpose of the 24 h recall was to enable the population under study to understand the correct filling of the instrument that quantified the average intake of vitamin A during the three-day food intake record. This was performed with a view to minimizing filling errors and improving the quality of information.

The three-day food intake record, including two days during the week and one on the weekend, was the method used to quantify the average intake of vitamin A in the study, as it was carried out at the time the food was being consumed. Thus, it was not based on the individual’s memory; instead, it measured current consumption and identified the types of food consumed and meal times. 

The responses obtained using the three-day food intake record were inserted and plotted in *Dietbox* software, which calculated the average daily VA intake according to the VA content in foods as published in the Brazilian Table of Food Composition [18], which is integrated into the *Dietbox* software. The portion size was evaluated using the Photo Atlas of Food Portion Sizes [19]. VA intake was compared with the daily intake values recommended by the Institute of Medicine [17]. The cutoff point adopted for the recommended dietary intake of VA was 700 μg/day of retinol equivalent.

The food consumption frequency questionnaire, based on a list of food sources of vitamin A recommended by the International Vitamin A Consultative Group (IVACG) [20], was used to determine how often these foods were consumed (daily, weekly, every two weeks, monthly, or never). The information was used during the nutritional guidance stage, and was provided to all the women who participated in the study. This guidance aimed to promote the necessary dietary changes, indicating the inclusion, greater frequency of consumption, or even the exclusion of foods in the nutritional guidance stage of our study.

One month after the first procedure, all women participating in the first stage of the study were scheduled to receive information about nutritional diagnosis, including VA intake, in addition to relevant dietary guidelines.

Only women who had a recommended dietary intake of VA according to the results of the dietary intake surveys used in the study were included (*n* = 231).

There was a loss of 31 women due to questionnaire completion errors and failure to submit the records. Two hundred women continued in the study and moved on to the third phase, where their body composition was assessed. Subsequently, the sample was divided into four groups according to BMI ranges: normal weight (NW), overweight (OW), obesity class I (OI), and obesity class II (OII). Participants were instructed to fast before blood collection. After this procedure, the women received the results of the biochemical tests, the diagnosis of the nutritional status of VA accompanied by appropriate nutritional guidance, and a prescription of VA supplementation (5000 IU/day of retinol palmitate for 12 weeks) for all those who had VAD.

### 2.3. Assessment of Body Variables

Weight and height were evaluated, and BMI was calculated by dividing body weight (in kilograms—kg) by height (in square meters—m^2^). The classification of this variable was performed considering the cut-off points proposed by the World Health Organization (WHO): normal weight, between 18.5 and 24.9 kg/m^2^; overweight, between 25.0 and 29.9 kg/m^2^; class I obesity, between 30.0 and 34.9 kg/m^2^; class II obesity, between 35.0 and 39.9 kg/m^2^; and class III obesity, ≥40 kg/m^2^ [1].

Abdominal adiposity was assessed using WC, WHtR HW, and BAI, while visceral adiposity was estimated by the VAI and LAP. The formulas and cut-off points used are presented below:WC is a widely used anthropometric parameter to assess abdominal fat. It was considered high if >88 cm [1].WHtR is applied to diagnose abdominal obesity and plays an important role in assessing the risk of cardiovascular events. It was calculated using WC (in cm) divided by Height (in cm), with a cut-off point >0.5 [21].HW is a marker for the simultaneous presence of WC and elevated serum triglyceride levels. It is a simple and practical indicator that can be used as a predictor of metabolic disease. It is characterized by the simultaneous presence of increased WC (≥80 cm) and elevated serum triglyceride (TG) levels (≥1.7 mmol/L) [22].BAI evaluates the percentage of body fat in adults. It is a method used to estimate body adiposity and is considered an alternative predictor of body fat in the absence of more complex techniques or more expensive methods. According to the formula: (hip circumference (cm) ÷ height (m) 1.5) − 18; the cut-off point is >33 [23].VAI can estimate the distribution of fat and the dysfunction of the visceral adipose tissue, resulting from a specific mathematical formula for each gender. According to the formula: (WC (cm) ÷ (36.58 + (BMI * 1.89) * (TG ÷ 0.81) * (1.52 ÷ HDL-c) for women, where TG and high-density lipoprotein cholesterol (HDL-c) are expressed in mmol/L, with a cut-off point >1 [24].LAP can represent lipotoxicity and may be a marker of abdominal adiposity that correlates with central fat accumulation. This index was calculated: (WC (cm) − 58) × (TG (mmol/L)). The cut-off point used was 37.9 [25].

### 2.4. Biochemical Measurements of Vitamin A

For the biochemical assessment of VA, a blood sample was obtained using venipuncture after 12 h fasting. Serum concentrations of retinol and β-carotene were quantified using High-Performance Liquid Chromatography with an ultraviolet detector (HPLC-UV); the following cut-off points were used for inadequate retinol (<1.05 µmol/L) and low β-carotene concentrations (<40 µg/dL) [26].

### 2.5. Other Biochemical Measurements

For other biochemical assessments, a blood sample was obtained using venipuncture after 12 h fasting. Laboratory tests were performed to assess the lipid profile (total cholesterol, TG, HDL-c, and low-density lipoprotein cholesterol [LDL-c]). The following concentrations were considered normal values: total cholesterol < 200 mg/dL; TG < 150 mg/dL; HDL-c > 50 mg/dL; and LDL-c < 150 mg/dL. The determinations of TG, total cholesterol, and HDL-c were performed using the colorimetric enzymatic method. Reagents for these biochemical evaluations were purchased from Labtest Diagnóstica S.A., Minas Gerais, Brazil. The LDL-c fraction was determined according to the Friedewald formula.

### 2.6. Statistical Analysis

The Kolmogorov–Smirnov test was used to test the normality of data and expressed as means and standard deviations for clinical, dietary, and biochemical variables. 

Analysis of variance (ANOVA) and Bonferroni’s test for multiple comparisons were used. Pearson’s correlation coefficient was applied for serum concentrations of retinol and β-carotene with body adiposity variables.

The significance level adopted was 5% (*p* < 0.05). Statistical analysis was performed using the Statistical Package for the Social Sciences for Windows version 26.

## 3. Results

The sample comprised 200 adult women who met the recommended dietary intake of VA (772.3 ± 59.9 μg/day—retinol equivalent). The mean age was 50.0 ± 5.6 years. Related to body composition of the total sample, the mean BMI, WC, WHtR, VAI, BAI and LAP were 28.1 ± 5.1 Kg/m^2^, 102.5 ± 21.0 cm, 0.6 ± 0.1, 4.4 ± 1.1, 31.2 ± 8.7 and 90.3 ± 55.1, respectively.

The general characteristics of the sample, according to BMI ranges, are described in Table 1.

According to the results, 59.5% of women belonged to the HW phenotype (*n* = 119). However, there was no relationship between the HW phenotype and serum concentrations of VA.

There was greater inadequacy of β-carotene when compared to retinol, where 55.5% of women (*n* = 101) showed deficiency in β-carotene (34.9 ± 13.8 µmol/L, *p* < 0.001), while 43.5% (*n* = 87) showed deficiency in retinol (0.71 ± 0.3 µmol/L, *p* < 0.001).

The mean serum concentrations of retinol and β-carotene of the groups are shown in Table 2. There was a significant decrease in serum concentrations of retinol and β-carotene as BMI increased. In addition, women classified as OI and OII had a deficiency in both retinol and β-carotene; women classified as OW had a deficiency in retinol. Only women classified as NW had adequate retinol and β-carotene levels.

Body adiposity parameters according to the nutritional status of retinol, β-carotene, and BMI groups are shown in Table 3. Greater inadequacy of Vitamin A, both retinol and β-carotene, can be observed as body adiposity increased, especially visceral adiposity.

A significant negative correlation was found between serum concentrations of retinol and β-carotene with body adiposity parameters (Table 4). 

## 4. Discussion

The increase in the prevalence of excess body weight among women suggests an important epidemiological scenario related to parity, sexual hormones, and metabolic health, favoring the redistribution of body fat, with emphasis on visceral fat, in addition to negatively influencing serum concentrations of micronutrients, such as vitamin A [2].

Obesity is not only an excessive accumulation of adipose tissue but may also be accompanied by low-grade inflammation with macrophage infiltration that forms a vicious cycle through a paracrine circle [27]. Hypertrophic adipocytes release excess saturated fatty acids and activate macrophages via the Toll-like receptor 4 signaling pathway. These macrophages secrete proinflammatory cytokines and react to hypertrophic adipocytes, inducing an inflammatory response by activating the nuclear factor kB pathway and promoting further release of free fatty acids, which aggravates obesity [28]. Some explanations can be proposed to account for the beneficial effect of carotenoids; carotenoids and their conversion products may affect the inflammatory and secretory profiles of adipose tissue by actively interacting with these pathways in adipocytes and adipose tissue macrophages [12].

In this context, the findings of the present study deserve attention as they demonstrate that excess body adiposity, especially visceral adiposity, may represent an important cause of VA depletion, even in the face of recommended intake. Furthermore, it reinforces that, in women with excess body adiposity, the nutritional needs of VA may be much higher than the current recommendations. This scenario is even worse if we consider the results of the most recent national survey conducted by the Family Budget Survey [16], carried out in urban and rural regions, that evaluated the adult food consumption of the Brazilian population. The survey demonstrated a very high prevalence of inadequate consumption of VA, affecting more than 80% of the adult population. Women have an even higher percentage (around 81.2%) [16].

Low concentrations of β-carotene were observed when compared to retinol. Since carotenoids, especially β-carotene, are considered precursors of retinoids, this indicates that, when their concentrations are reduced, retinoid synthesis is also reduced, thereby affecting the body’s ability to prevent weight gain and obesity [29]. β-carotene inhibits adipogenesis through the production of β-apo-140-carotene and the suppression of peroxisome proliferator-activated receptor type alpha (PPARα), peroxisome proliferator-activated receptor type gamma (PPARγ), and retinoid X receptor (RXR), in addition to the production of all-trans retinoic acid [30]. This study showed that adipocyte differentiation followed by a change in all-trans retinoic acid signaling, in mature adipocytes, enabled all-trans retinoic acid to activate both retin-oic acid receptors and PPARβ/δ, increasing lipolysis and depleting lipid stores. Therefore, these results indicate that improvement of obesity and insulin resistance by retinoic acid is widely mediated by PPARβ/δ and is further enhanced by activation of all-trans retinoic acid receptors [31].

Studies suggest that pro-VA carotenoids, including β-carotene, assist in the control of energy homeostasis by modulating the production of leptin and inflammatory cytokines [32,33]. Furthermore, in an experimental study [30], β-carotene was associated, in a dose-dependent manner, with increased expression of brown adipocyte uncoupling protein 1 (UCP-1), that dissipates energy from ATP synthesis for heat generation. 

The use of different indices and parameters to evaluate body adiposity deserves to be highlighted, especially for those related to visceral fat, which, in excess, leads to an inflammatory process, the presence of insulin resistance (IR), and adiponectin dysregulation. This study also showed that serum concentrations of retinol and β-carotene were reduced as body adiposity, especially visceral adiposity, increased when evaluated by WC, WHtR, VAI, and LAP. This finding can be explained, in part, by the greater demand for nutrients with antioxidant action. Once the visceral adipose tissue (VAT) expands, it becomes inflamed due to the infiltration of macrophages into the hypertrophied adipocytes, leading to increased production of inflammatory cytokines, including interleukin-6 (IL-6), tumor necrosis factor-alpha (TNF-α), and interleukin-1β (IL-1β); the reduced production of anti-inflammatory cytokines, such as adiponectin [34], consequently increases the release of reactive oxygen species, giving rise to oxidative stress (OS) [35,36].

These indices have shown promise because they have a good ability to predict adverse health outcomes. The way fat deposits are distributed, particularly in VAT, is considered the major contributor to metabolic risk, and results in different metabolic impacts more strongly associated with OS, inflammatory profile, IR, and higher mortality in general; this is more often seen in women [37,38,39,40,41].

Our findings are in line with the studies developed by Bento et al. (2018) [14] and Kabat et al. (2016) [42] on the inverse relationship between body adiposity and serum concentrations of retinol and β-carotene. However, the use of different indices of body adiposity measurement demonstrated that body fat distribution is a key issue regarding the impact of adiposity on serum concentrations of VA.

Given the results, it is important to emphasize the role of VA metabolites in the activity of differentiation and functionality in adipose tissue, such as inhibition of cell differentiation in adipocyte culture. The action of retinoic acid in inducing UCP-1 expression and the consequent activation of thermogenesis in brown adipose tissue and the browning of white adipose tissue were well established [12]. In addition, retinoic acid is also associated with increased lipolysis in white adipose tissue via PPAR-y, decreased RXR expression, improved oxidative metabolism [32], increased lipolysis in adipocytes, decreased leptin and resistin expression, activation of thermogenesis by UCPs, and reduced cell differentiation of pre-adipocytes into adipocytes, thus reducing lipid storage capacity in adipocytes due to the action of the enzyme monooxygenase-1 β, β-carotene [43]. 

In this scenario, increased body adiposity can trigger VA depletion via the increased OS, inflammation, and, in some cases, in adipocyte sequestration [30]. Additionally, a deficiency of this vitamin can activate additional metabolic pathways associated with increased body adiposity. The way body fat was distributed, especially high visceral adiposity, showed an association with compromised nutritional status regardless of the recommended dietary intake [44].

One limitation of this study is related to the assessment of food intake, the same limitations found in most studies based on self-reported recall, in particular the underreporting of intake. However, we see no reason to believe that underreporting could have been different among the groups we evaluated. 

The findings of the present study reinforce the inclusion of WC, WHtR, VAI, and LAP as useful and complementary tools for the assessment of body adiposity since they provide additional information about the distribution of body fat and the relationship of this distribution with the nutritional status of VA. We suggest the development of further studies with the inclusion of inflammatory markers, aiming at greater detailing about the relationship between VA and body fatness. Such information may support the development of control and prevention strategies for two major public health problems: obesity and VAD. 

## 5. Conclusions

This study demonstrated an association between inadequate serum concentrations of retinol and β-carotene with excess body adiposity in women, even with the recommended intake of this vitamin. It was possible to observe the impaired nutritional status of VA by assessing body adiposity through the different indices and parameters used; these showed that the site of fat distribution is an impact factor on the serum concentrations of this vitamin.

Considering the importance of VA in the metabolic regulation of adipose tissue, it is important to monitor its serum concentrations as well as to assess body fat distribution by including body adiposity assessment parameters such as WC, WHtR, VAI, and LAP.

## Figures and Tables

**Figure 1 biomedicines-11-00991-f001:**
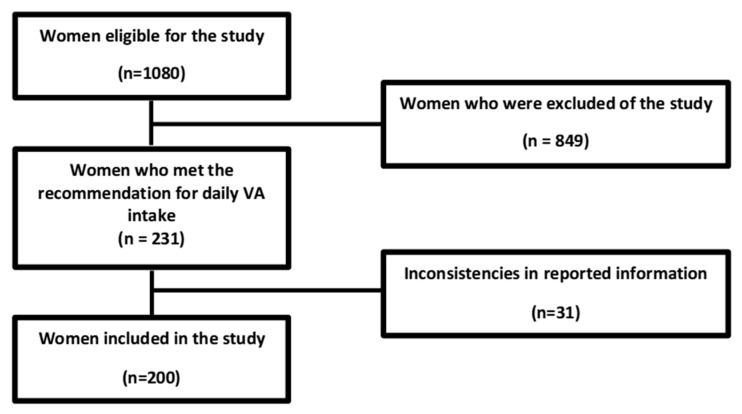
Flowchart describing the recruitment of the study participants.

**Table 1 biomedicines-11-00991-t001:** General characteristics of the sample according to BMI ranges.

Variables	NW (*n* = 80)	OW (*n* = 40)	OI (*n* = 68)	OII (*n* = 12)	*p*-value
BMI (kg/m^2^)	22.8 ± 1.1	27.3 ± 1.2	33.1 ± 0.9	37.7 ± 0.9	<0.001
Age (years)	48.2 ± 5.7	50.8 ± 5.6	50.8 ± 5.1	54.3 ± 3.7	<0.001
WC (cm)	79.4 ± 6.1	112.4 ± 10.3	120.6 ± 10.2	121.1 ± 8.0	<0.001
WHtR	0.5 ± 0.0	0.7 ± 0.1	0.7 ± 0.1	0.7 ± 0.0	<0.001
VAI	3.4 ± 0.7	4.8 ± 0.7	5.1 ± 0.8	5.3 ± 0.4	<0.001
BAI	22.4 ± 4.3	36.6 ± 6.3	37.3 ± 5.0	37.7 ± 3.6	<0.001
LAP	33.8 ± 12.2	101.2 ± 22.2	137.2 ± 35.8	165.1 ± 30.4	<0.001
Retinol equivalent (μg/day)	795.2 ± 49.9	781.5 ± 34.5	745.8 ± 72.8	740.1 ± 32.3	0.243

ANOVA. Mean and standard deviation. BMI: Body Mass Index; WC: Waist Circumference; WHtR: Waist-to-Height Tatio; VAI: Visceral Adiposity Index; BAI: Body Adiposity Index; LAP: Lipid Accumulation Product; NW: Normal Weight; OI: Obesity Class I; OII: Obesity Class II; OW: Overweight.

**Table 2 biomedicines-11-00991-t002:** Mean and standard deviation of serum concentrations of retinol and β-carotene according to BMI groups.

BMI Groups	Retinol (μmol/L)	β-Carotene (μg/dL)
NW (*n* = 80)	1.3 ± 0.2	61.2 ± 12.1
OW (*n* = 40)	1.0 ± 0.3	43.5 ± 5.9
OI (*n* = 68)	0.8 ± 0.3	35.9 ± 4.3
OII (*n* = 12)	0.7 ± 0.2	32.0 ± 0.9
*p*-value	<0.001	<0.001

Mean and standard deviation. ANOVA test: BMI: Body Mass Index; NW: Normal Weight; OI: Obesity Class I; OII: Obesity Class II; OW: Overweight.

**Table 3 biomedicines-11-00991-t003:** Body adiposity parameters according to Retinol nutritional status, β-carotene in BMI groups.

BMI Groups	WC (cm)	WHtR	VAI	BAI	LAP
NW (*n* = 80)	Retinol LConc 88.5 ± 14.5 Adq 78.6 ± 4.3	Retinol LConc 0.6 ± 0.1 Adq 0.5 ± 0.0	Retinol LConc 3.4 ± 0.9 Adq 3.4 ± 0.7	Retinol LConc 27.4 ± 5.8 Adq 22.0 ± 3.9	Retinol LConc 55.3 ± 31.0 Adq 32.1 ± 7.3
	β-carotene LConc 83.3 ± 11.1 Adq 78.7 ± 4.5	β-carotene LConc 0.5 ± 0.1 Adq 0.5 ± 0.0	β-carotene LConc 3.4 ± 0.8 Adq 3.4 ± 0.7	β-carotene LConc 24.6 ± 5.4 Adq 22.0 ± 4.0	β-carotene LConc 43.4 ± 24.3 Adq 32.1 ± 7.6
	*p*-value < 0.001	*p*-value < 0.001	*p*-value < 0.001	*p*-value < 0.001	*p*-value < 0.001
	Retinol LConc 111.2 ± 11.3 Adq 113.2 ± 9.6	Retinol LConc 0.7 ± 0.1 Adq 0.7 ± 0.0	Retinol LConc 5.1 ± 0.6 Adq 4.6 ± 0.7	Retinol LConc 38.9 ± 7.8 Adq 34.8 ± 4.4	Retinol LConc 104.3 ± 27.4 Adq 98.9 ± 17.6
OW (*n* = 40)	β-carotene LConc 111.0 ± 12.8 Adq 113.6 ± 7.4 *p*-value < 0.001	β-carotene LConc 0.7 ± 0.1 Adq 0.7 ± 0.0 *p*-value < 0.001	β-carotene LConc 5.1 ± 0.7 Adq 4.6 ± 0.7 *p*-value < 0.001	β-carotene LConc 38.6 ± 8.1 Adq 34.7 ± 3.3 *p*-value < 0.001	β-carotene LConc 102.8 ± 29.0 Adq 99.7 ± 14.0 *p*-value < 0.001
OI (*n* = 68)	Retinol LConc 121.5 ± 11.2 Adq 117.8 ± 4.9	Retinol LConc 0.7 ± 0.1 Adq 0.7 ± 0.0	Retinol LConc 5.3 ± 0.8 Adq 4.4 ± 0.4	Retinol LConc 37.7 ± 5.4 Adq 36.0 ± 3.4	Retinol LConc 145.8 ± 36.6 Adq 109.4 ± 9.4
	β-carotene LConc 121.0 ± 10.9 Adq 118.5 ± 3.4	β-carotene LConc 0.7 ± 0.1 Adq 0.7 ± 0.0	β-carotene LConc 5.3 ± 0.8 Adq 4.4 ± 0.4	β-carotene LConc 37.3 ± 5.3 Adq 37.2 ± 2.9	β-carotene LConc 142.2 ± 36.5 Adq 108.4 ± 7.3
	*p*-value < 0.001	*p*-value < 0.001	*p*-value < 0.001	*p*-value < 0.001	*p*-value < 0.001
OII (*n* = 12)	Retinol LConc 121.1 ± 8.0	Retinol LConc 0.7 ± 0.0	Retinol LConc 5.3 ± 0.4	Retinol LConc 37.7 ± 3.6	Retinol 165.1 ± 30.4
	β-carotene LConc 121.1 ± 8.1	β-carotene LConc 0.7 ± 0.0	β-carotene LConc 5.3 ± 0.4	β-carotene LConc 37.7 ± 3.6	β-carotene LConc 165.1 ± 30.4
	*p*-value < 0.001	*p*-value < 0.001	*p*-value < 0.001	*p*-value < 0.001	*p*-value < 0.001

Mean and standard deviation. ANOVA test: Adq: Adequacy; BMI: Body Mass Index; LConc: low concentration; NW: Normal Weight; OI: Obesity Class I; OII: Obesity Class II; OW: Overweight.

**Table 4 biomedicines-11-00991-t004:** Correlation of serum concentrations of retinol and β-carotene with body adiposity parameters.

Body Adiposity Parameters	Retinol (μmol/L)		β-Carotene (μg/dL)	
	r	*p*	r	*p*
BMI (Kg/m^2^)	−0.65	<0.001	−0.76	<0.001
WC (cm)	−0.71	<0.001	−0.77	<0.001
WHtR	−0.72	<0.001	−0.73	<0.001
VAI	−0.73	<0.001	−0.68	<0.001
BAI	−0.70	<0.001	−0.71	<0.001
LAP	−0.81	<0.001	−0.78	<0.001

Pearson’s Correlation. BMI: Body Mass Index; BAI: Body Adiposity Index; LAP: Lipid Accumulation Product; VAI: Visceral Adiposity Index; WC: Waist Circumference; WHtR: Waist-to-Height Ratio.

## Data Availability

The data used to support the findings of this study are available from the corresponding author upon request.

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
