# Peer review of "Vitamin A Deficiency and Its Association with Visceral Adiposity in Women"

_biomedicines, 2023, doi:10.3390/biomedicines11030991_

Round 1

Reviewer 1 Report

The manuscript entitled „ Vitamin A deficiency and its association with visceral adiposity in women” presents interesting issue, but some problems should be corrected.

General:

English language in the whole manuscript should be polished – e.g. “women in different BMI classes, seen at the Nutrition Service” (seen by whom?)

Abstract:

The Abstract should be prepared according to the instructions for authors (without subheadings)

different parameters and indexes” should be clearly listed

Introduction:

Authors should prepare this section not only to be interesting for Brazilian readers, but to be interesting for international readers. If Authors prepare their manuscript only for their national readers, they should publish it in some national journal. So, Authors should present here international data from various countries, not only the Brazilian ones.

Materials and Methods:

Lines 82-84 – “This section may be divided by subheadings. It should provide a concise and precise description of the experimental results, their interpretation, as well as the experimental conclusions that can be drawn” – should be removed

Instead of “≥ 20 ≤ 49 years”, Authors should specify it as age <20; 49>

It is hard to guess how did Authors assess diet vitamin A dietary intake, as they declared as “a food intake survey was performed comprising a 24-hour recall using three food records, two records during the week and one record at the weekend, and a food frequency questionnaire”, so they should declare how did they combine vitamin A data from dietary record and food frequency questionnaire

The flowchart is improperly prepared, as Authors indicated excluded individuals (231) as those from which were chosen included ones (200). And they didi not indicate what did happen with 1080 – 231 = 849 individuals

It seems that Authors did not verify the normality of distribution and they treated all the variables as normally distributed.

Authors should (1) verify the normality of distribution, (2) for normally distributed data present mean and SD values, but for the other distributions – present median, min and max values, (3) apply adequate statistical tests, that are based on the distribution.

Results:

The representativeness of the studied group should be verified here

It seems that Authors did not verify the normality of distribution and they treated all the variables as normally distributed.

Authors should (1) verify the normality of distribution, (2) for normally distributed data present mean and SD values, but for the other distributions – present median, min and max values, (3) apply adequate statistical tests, that are based on the distribution.

The dietary intake of vitamin A should be presented as a characteristics of the studied group and compared between subgroups

Discussion:

The representativeness of the studied group should be discussed here

Author Contributions:

Authors should properly define the contributions – e.g. what do Authors mean by “validation” if they did not conduct any validation in their study? There are similar doubts in case of “resources” and “data curation”

It seems that contribution of AM and CB was only minor and they did not participate in preparing manuscript. There is a serious risk of a guest authorship procedure which is forbidden. In such case (if they did not participate in manuscript preparation in any way) they should be rather presented in Acknowledgements Section and not be indicated as authors of the study.

Author Response

LETTER TO REVIEWERS

Biomedicines – Manuscript ID 2199431

Title: “Vitamin A deficiency and its association with visceral adiposity in women”

Reviewer 1

The manuscript entitled „ Vitamin A deficiency and its association with visceral adiposity in women” presents interesting issue, but some problems should be corrected.

1.General:

English language in the whole manuscript should be polished – e.g. “women in different BMI classes, seen at the Nutrition Service” (seen by whom?)

Reply: The phrase was reformulated for better understanding.

2.Abstract:

The Abstract should be prepared according to the instructions for authors (without subheadings)

Reply: The authors removed all subheadings according to the instructions.

“different parameters and indexes” should be clearly listed

Reply: The authors added all parameters and indexes that were used in the study.

3.Introduction:

Authors should prepare this section not only to be interesting for Brazilian readers, but to be interesting for international readers. If Authors prepare their manuscript only for their national readers, they should publish it in some national journal. So, Authors should present here international data from various countries, not only the Brazilian ones.

Reply: Dear reviewer, the introduction was developed to all readers nationality, the authors did not give much attention for Brazilians women, only mentioned in the first paragraph the epidemiology data about excess of weight in this population to contextualize.

“The prevalence of excess adiposity is higher among women, and in the world population, about 40% of adult women are classified with overweight (OW) and 15% with obesity (O) [1]. In Brazil, more than half (50.5%) of women are classified with overweight and 19.6% with obesity [2].”

4.Materials and Methods:

Lines 82-84 – “This section may be divided by subheadings. It should provide a concise and precise description of the experimental results, their interpretation, as well as the experimental conclusions that can be drawn” – should be removed

Reply: The authors removed this sentence.

5.Instead of “≥ 20 ≤ 49 years”, Authors should specify it as age <20; 49>

Reply: The authors substituted the range of age, according to contribution from the reviewer.

6.It is hard to guess how did Authors assess diet vitamin A dietary intake, as they declared as “a food intake survey was performed comprising a 24-hour recall using three food records, two records during the week and one record at the weekend, and a food frequency questionnaire”, so they should declare how did they combine vitamin A data from dietary record and food frequency questionnaire

Reply: Dear reviewer for better understanding of the manuscript methodology, the authors reformulated the writing, clarifying all the details about assessment of vitamin A intake.

7.The flowchart is improperly prepared, as Authors indicated excluded individuals (231) as those from which were chosen included ones (200). And they did not indicate what did happen with 1080 – 231 = 849 individuals

Reply: The authors re-formulated the flowchart for better understanding.

8.It seems that Authors did not verify the normality of distribution and they treated all the variables as normally distributed.

9.Authors should (1) verify the normality of distribution, (2) for normally distributed data present mean and SD values, but for the other distributions – present median, min and max values, (3) apply adequate statistical tests, that are based on the distribution.

Reply: The Research Center statistician developed all the adequate statistical tests and was verified the normality of distribution, and cause of this the data were presented as mean and SD values. To become clear the analysis, the authors inserted this evaluation into the Statistical Analysis.

  1. Results:

The representativeness of the studied group should be verified here

It seems that Authors did not verify the normality of distribution and they treated all the variables as normally distributed.

Reply: The Research Center statistician developed all the adequate statistical tests and was verified the normality of distribution, and cause of this the data were presented as mean and SD values. To become clear the analysis, the authors inserted this evaluation into the Statistical Analysis.

Authors should (1) verify the normality of distribution, (2) for normally distributed data present mean and SD values, but for the other distributions – present median, min and max values, (3) apply adequate statistical tests, that are based on the distribution.

Reply: The Research Center statistician developed all the adequate statistical tests and was verified the normality of distribution, and cause of this the data were presented as mean and SD values. To become clear the analysis, the authors inserted this evaluation into the Statistical Analysis.

The dietary intake of vitamin A should be presented as a characteristic of the studied group and compared between subgroups

Reply: Thank you for your suggestion, however the aim of this manuscript was to evaluate the relationship between serum concentrations of VA and body adiposity, especially visceral adiposity, by using different parameters and indexes in women with the recommended intake of this vitamin, due to the scarcity of studies about this subject.

  1. Discussion:

The representativeness of the studied group should be discussed here

Reply: The authors inserted the representativeness of the studied group into the discussion.

  1. Author Contributions:

Authors should properly define the contributions – e.g. what do Authors mean by “validation” if they did not conduct any validation in their study? There are similar doubts in case of “resources” and “data curation”

Reply: The term “validation” was used to state that the food consumption frequency questionnaire used in the study was validated and recommended by the International Vitamin A Consultative Group (IVACG/WHO). All the required resources, no financial, to develop all the phases of the study were provided by C.B. and A.M., as well, data curation.

It seems that contribution of AM and CB was only minor and they did not participate in preparing manuscript. There is a serious risk of a guest authorship procedure which is forbidden. In such case (if they did not participate in manuscript preparation in any way) they should be rather presented in Acknowledgements Section and not be indicated as authors of the study.

Reply: All authors mentioned in this manuscript played an important role in its development. AM and CB were the researchers who carried out the consultations of the women participating in the research, in its most different phases, as well as participating in all the meetings to discuss the findings and forward them to the elaboration of the manuscript. Therefore, they cannot be considered only as research guests.

Thank you for all consideration and time to review our manuscript.

Best regards,

Reviewer 2 Report

Aim of this cross-sectional study is to evaluate serum levels of vitamin A and association with body adiposity in women with recommended intake of vitamin A. 

Scientific literature proposes that vitamin A could affect obesity development and the development of obesity-related diseases including insulin resistance, type 2 diabetes, hepatic steatosis and steatohepatitis, and cardiovascular disease. 

This paper must be thoroughly reviewed before possible publication

1. The discussion of this paper is dull. The pathogenetic mechanisms that could explain the results need to be investigated in depth. For instance, on a molecular level it has long been appreciated that retinoic acid can inhibit adipogenesis. However, in recent years new research has emerged pointing to new molecular pathways for retinoid functioning. Results from these studies suggest that alterations of retinoid metabolism affect the activity of the master regulator PPARγ 

2. Adipose tissue has an active vitamin A metabolism but it is also a stores vitamin A. Could vitamin A sequestration in adipose tissue explain the results? The more body fat increases, the more vitamin A is sequestered and diluted

3. The lack of body composition assessment and the absence of diet assessment in this study are  very serious shortcomings.

4. Plagiarism is high (please see the attachment)

Author Response

LETTER TO REVIEWERS

Biomedicines – Manuscript ID 2199431

Title: “Vitamin A deficiency and its association with visceral adiposity in women”

Reviewer 2

Comments and Suggestions for Authors

Aim of this cross-sectional study is to evaluate serum levels of vitamin A and association with body adiposity in women with recommended intake of vitamin A. 

Scientific literature proposes that vitamin A could affect obesity development and the development of obesity-related diseases including insulin resistance, type 2 diabetes, hepatic steatosis and steatohepatitis, and cardiovascular disease. 

This paper must be thoroughly reviewed before possible publication

1.The discussion of this paper is dull. The pathogenetic mechanisms that could explain the results need to be investigated in depth. For instance, on a molecular level it has long been appreciated that retinoic acid can inhibit adipogenesis. However, in recent years new research has emerged pointing to new molecular pathways for retinoid functioning. Results from these studies suggest that alterations of retinoid metabolism affect the activity of the master regulator PPARγ.

Reply: The authors agreed with the reviewer that is necessary more deep investigations, mainly genetic ones. But we could explain some metabolic mechanisms related to ours results. And to enrich the discussion, the authors inserted more information about PPAR pathway.

  1. Adipose tissue has an active vitamin A metabolism but it is also a stores vitamin A. Could vitamin A sequestration in adipose tissue explain the results? The more body fat increases, the more vitamin A is sequestered and diluted

Reply: The authors could observe that some studies have described that the greater the adiposity, the greater the storage of carotenoids in adipose tissue, since it is a fat- soluble nutrient (Chung HY et al, 2009; Nuss ET et al, 2017). However, recent studies have presented data that contradict, in part, this affirmative about it, describing findings that observed lower concentrations of β-carotene in the adipose tissue of individuals with obesity compared to normal weight ones (Östh M et al, 2014; Harari A et al, 2020). Thus, although there is a greater uptake of carotenoids by adipose tissue, they play a local antioxidant role in combating inflammation and oxidative stress that occur in these tissues, in addition to acting as retinol precursors (mainly retinoic acid), which may result at lower concentrations at both systemic and tissue levels among individuals with high body adiposity (Harari A et al, 2020).

Thus, even though there is a greater uptake of carotenoids by adipose tissue, they play a local antioxidant role in fighting inflammation and oxidative stress that occur in these tissues, besides acting as precursors for retinol (mainly retinoic acid), which may result in lower concentrations both at systemic and tis- sue levels among individuals with high body adiposity (Harari A et al, 2020; Ribamar A et al, 2022).

References:

Nuss ET, Valentine AR, Zhang Z, Lai HJ, Tanumihardjo SA. Serum carotenoid interactions in premenopausal women reveal α-carotene is negatively im- pacted by body fat. Exp Biol Med Maywood NJ 2017;242:1262–70. doi: 10.1177/ 1535370217706962 .

Harari A, Coster ACF, Jenkins A, Xu A, Greenfield JR, Harats D, et al. Obesity and insulin resistance are inversely associated with serum and adipose tis- sue carotenoid concentrations in adults. J Nutri 2020;150:38–46. doi: 10.1093/ jn/nxz184 .

Chung HY, Ferreira ALA, Epstein S, Paiva SAR, Castaneda-Sceppa C, Johnson EJ. Site-specific concentrations of carotenoids in adipose tissue: relations with di- etary and serum carotenoid concentrations in healthy adults. Am J Clin Nutr 20 09;90:533–9. doi: 10.3945/ajcn.20 09.27712.

Östh M, Öst A, Kjolhede P, Strålfors P. The concentration of β-carotene in hu- man adipocytes, but not the whole-body adipocyte stores, is reduced in obe- sity. PLoS ONE 2014;9. doi: 10.1371/journal.pone.0085610.

Ribamar A, Cruz S, Bento C, Ramalho A. Visceral and body adiposity are negatively associated with vitamin A nutritional status independently of Body Mass Index and recommended intake of vitamin A in Brazilian Women. J Nutr Biochem. 2022 Nov;109:109-120. doi: 10.1016/j.jnutbio.2022.109120.

  1. The lack of body composition assessment and the absence of diet assessment in this study are very serious shortcomings.

Reply: Dear reviewer, the manuscript has a huge body composition assessment through weight (Kg), height (cm), Body mass index [(BMI) - Kg/m²], waist circumference (WC), waist-to-height ratio (WHtR), hypertriglyceridemic waist (HW), body adiposity index (BAI), while visceral adiposity was estimated by the visceral adiposity index (VAI) and lipid accumulation product (LAP).

  • WC is a widely used anthropometric parameter to assess abdominal fat, estimate visceral adiposity and is considered an independent risk factor for cardiometabolic complications such as insulin resistance, Type 2 Diabetes Mellitus and atherosclerosis
  • WHtR is applied to diagnose abdominal obesity, playing an important role in assessing the risk of cardiovascular events. It has been suggested as an alternative index for assessing obesity, being superior as an indicator of mortality risk and a risk factor for cardiometabolic diseases, when compared to BMI and WC.
  • HW is a marker for the simultaneous presence of WC and elevated serum triglyceride levels. It is a simple and practical indicator that can be used as a predictor of the atherogenic metabolic triad (presence of hyperinsulinemia, elevated levels of apolipoprotein B and increased concentrations of small and dense particles of low-density lipoprotein cholesterol [LDL-c]) and of risk of metabolic diseases.
  • The VAI, resulting from a specific mathematical formula for each gender, is capable of estimating the distribution of fat and dysfunction of the visceral adipose tissue. It considers anthropometric and biochemical variables that are easy to obtain and interpret, namely WC (cm), BMI (kg/m²), triglycerides (mmol/L) and high-density lipoprotein cholesterol HDL-c (mmol/L). It has a positive correlation with insulin resistance and a negative correlation with cardiovascular risk, having already been identified as an independent predictor of cardiovascular risk in 10 years. In addition, it has a good correlation with gold standard methods, such as DXA.
  • The BAI evaluates the percentage of body fat in adults, being calculated from the hip circumference and height of the individual, being a method to estimate body adiposity and considered an alternative predictor of body fat in the absence of more complex techniques or methods more expensive, with the advantage of presenting a strong significant correlation with the percentage of fat estimated by skinfolds and bioimpedance.
  • LAP can represent lipotoxicity and may be a marker of abdominal adiposity that correlates with central fat accumulation. Regarding body composition, this index showed a strong correlation with trunk fat and fat mass/height and moderately associated with total body fat, total trunk/leg fat mass (%), evaluated by DEXA.

Therefore, the use of different indices and parameters to evaluate body adiposity deserves was a highlight of the study, especially for evaluation related to visceral fat, which in excess leads to an inflammatory process and the presence of insulin resistance.

References:

WORLD HEALTH ORGANIZATION (WHO). Obesity and Overweight. Revised February of 2018. Available from:  http://www.who.int/mediacentre/factsheets/fs311/en/ Accessed: March 26, 2018.

Zeng Q, He Y, Dong S, Zhao X, Chen Z, Song Z, Chang G, Yang F, Wang Y. Optimal cut-off values of BMI, waist circumference and waist: height ratio for defining obesity in Chinese adults. Br J Nutr. 2014;112(10):1735–1744. doi: 10.1017/S0007114514002657.

Freitas, Roberta Souza, Fonseca, Maria de Jesus Mendes da, Schmidt, Maria Inês, Molina, Maria del Carmen Bisi, & Almeida, Maria da Conceição Chagas de. (2018). Fenótipo cintura hipertrigliceridêmica: fatores associados e comparação com outros indicadores de risco cardiovascular e metabólico no ELSA-Brasil. Cadernos de Saúde Pública, 34(4), e00067617. Epub March 29, 2018.https://dx.doi.org/10.1590/0102-311x00067617.

Bergman RN, Stefanovski D, Buchanan TA, et al. A Better Index of Body Adiposity. Obesity (Silver Spring, Md). 2011;19(5):1083-1089. doi:10.1038/oby.2011.38.

Amato MC, Giordano C, Galia M, et al. Visceral Adiposity Index: A reliable indicator of visceral fat function associated with cardiometabolic risk. Diabetes Care. 2010;33(4):920-922. doi:10.2337/dc09-1825.

Vieira JN, Braz MAD, Gomes FO, Silva PRD, Santos OTM, Rocha IMGD, Sousa IM, Fayh APT. Cardiovascular risk assessment using the lipid accumulation product index among primary healthcare users: a cross-sectional study. Sao Paulo Med J. 2019 Jul 15;137(2):126-131. doi: 10.1590/1516-3180.2018.0293240119. PMID: 31314872.

Reply: Regarding the assessment of vitamin A consumption, the 24 hours recall, food records and the frequency of consumption validated and recommended by the IVACG/WHO were used. The authors assumed the limitation of the assessment of food consumption in the study. However, the same limitation based on self-reported recall was found in most studies published in indexed journals, we see no reason to believe that underreporting could have been different between the groups we evaluated.

  1. Plagiarism is high (please see the attachment)

Reply: The authors have checked the document for plagiarism and made a great revision in a manuscript to reach the acceptable percentage.

Thank you for all consideration and time to review our manuscript.

Best regards,

Reviewer 3 Report

Comments to Authors              

            This study showed: a) an association between inadequate serum concentrations of retinol and β-carotene with excess body adiposity in women, even with the recommended intake of this vitamin; b) it was possible to observe the impaired nutritional status of VA by assessing body adiposity by the different indices and parameters used, which showed that the site of fat distribution as an impact factor on the serum concentrations of this vitamin.

          Authors are kindly requested to emphasize the current concepts about these issues in the context of recent knowledge and the available literature. This articles should be quoted in the References list.

References

1.      Physical Activity, Adiposity, and Serum Vitamin D Levels in Healthy Women: The Cooper Center Longitudinal Study. J Womens Health (Larchmt). 2022; 31 (7): 957-964. doi:10.1089/jwh.2021.0402.

2.      Visceral and body adiposity are negatively associated with vitamin A nutritional status independently of Body Mass Index and recommended intake of vitamin A in Brazilian Women. J Nutr Biochem. 2022; 109: 109120. doi:10.1016/j.jnutbio.2022.109120.

Author Response

LETTER TO REVIEWERS

Biomedicines – Manuscript ID 2199431

Title: “Vitamin A deficiency and its association with visceral adiposity in women”

Reviewer 3

Comments to Authors              

 This study showed: a) an association between inadequate serum concentrations of retinol and β-carotene with excess body adiposity in women, even with the recommended intake of this vitamin; b) it was possible to observe the impaired nutritional status of VA by assessing body adiposity by the different indices and parameters used, which showed that the site of fat distribution as an impact factor on the serum concentrations of this vitamin.

Authors are kindly requested to emphasize the current concepts about these issues in the context of recent knowledge and the available literature. This articles should be quoted in the References list.

References

  1. Physical Activity, Adiposity, and Serum Vitamin D Levels in Healthy Women: The Cooper Center Longitudinal Study. J Womens Health (Larchmt). 2022; 31 (7): 957-964. doi:10.1089/jwh.2021.0402.
  2. Visceral and body adiposity are negatively associated with vitamin A nutritional status independently of Body Mass Index and recommended intake of vitamin A in Brazilian Women. J Nutr Biochem. 2022; 109: 109120. doi:10.1016/j.jnutbio.2022.109120.

Reply: Dear reviewer, thank you for your suggestions. The first article referred to vitamin D, that even being a fat-soluble vitamin, has a different metabolic pathway compared with vitamin A. And also, this article is related to physical activity, that in our manuscript was not evaluated.

The second manuscript was developed in our Research Center, and we could give more information in our discussion with data from it.

Thank you for all consideration and time to review our manuscript.

Best regards,

Round 2

Reviewer 1 Report

The manuscript entitled „ Vitamin A deficiency and its association with visceral adiposity in women” presents interesting issue, but some problems should be corrected.

Introduction:

Authors should prepare this section not only to be interesting for Brazilian readers, but to be interesting for international readers. If Authors prepare their manuscript only for their national readers, they should publish it in some national journal. So, Authors should present here international data from various countries, not only the Brazilian ones. The epidemiological data are needed.

Materials and Methods:

It is still hard to guess how did Authors assess diet vitamin A dietary intake, as they declared as “a food intake survey was performed comprising a 24-hour recall using three food records, two records during the week and one record at the weekend, and a food frequency questionnaire”, so they should clearly declare how did they combine vitamin A data from dietary record and food frequency questionnaire. They indicated that they calculated average intake, but we still do not know how – if it was mean of 3 vitamin A values, it is improper approach, so it should be clearly indicated. It is crucial for the studied issue.

Results:

The representativeness of the studied group should be verified here – characteristics of the studied group while compared with the general population

The dietary intake of vitamin A should be presented as a characteristics of the studied group and compared between subgroups

Discussion:

The representativeness of the studied group should be discussed here

Author Contributions:

Authors should properly define the contributions – e.g. what do Authors mean by “validation” if they did not conduct any validation in their study? There are similar doubts in case of “resources” and “data curation”. Only actions associated with conducting this specific study and preparing manuscript should be indicated here.

It seems that contribution of AM and CB was only minor and they did not participate in preparing manuscript. There is a serious risk of a guest authorship procedure which is forbidden. In such case (if they did not participate in manuscript preparation in any way) they should be rather presented in Acknowledgements Section and not be indicated as authors of the study.

Author Response

LETTER TO REVIEWERS

Biomedicines – Manuscript ID 2199431

Title: “Vitamin A deficiency and its association with visceral adiposity in women”

Reviewer 1

The manuscript entitled “Vitamin A deficiency and its association with visceral adiposity in women” presents interesting issue, but some problems should be corrected.

Introduction:

Authors should prepare this section not only to be interesting for Brazilian readers, but to be interesting for international readers. If Authors prepare their manuscript only for their national readers, they should publish it in some national journal. So, Authors should present here international data from various countries, not only the Brazilian ones. The epidemiological data are needed.

Reply: The authors inserted international epidemiological data.

Materials and Methods:

It is still hard to guess how did Authors assess diet vitamin A dietary intake, as they declared as “a food intake survey was performed comprising a 24-hour recall using three food records, two records during the week and one record at the weekend, and a food frequency questionnaire”, so they should clearly declare how did they combine vitamin A data from dietary record and food frequency questionnaire. They indicated that they calculated average intake, but we still do not know how – if it was mean of 3 vitamin A values, it is improper approach, so it should be clearly indicated. It is crucial for the studied issue.

Reply: The authors inserted more information about assessment of vitamin A dietary intake in the methodology. And to become clear all process, we inform below all details:

The 24-hour recall was used as one of the nutritional assessment components in the present study. Through its use, the consumption of vitamin A on the day before the interview was investigated, considering the preparation, information on weight and size of portions, in grams, milliliters and in household measures. This was considered an important moment in our work, mainly because it was possible to carry out training before completing the instrument that evaluated the average consumption of vitamin A, the three-day food intake record, aiming to minimize filling errors and, consequently, improve the information quality.

The three-day food intake record, including one day on the weekend, was the method used to quantify the average intake of vitamin A in the study, as it was carried out at the time the food was consumed. Thus, it is not based on the individual's memory, it measures current consumption, identifies types of preparations consumed and mealtimes. The responses obtained using the three-day food intake record were double-coded and entered into a personalized spreadsheet that calculated the average daily VA intake according to the VA content in foods as published in the Brazilian Table of Food Composition.

The food consumption frequency questionnaire, based on a list of food sources of vitamin A recommended by the IVACG, was used to find out how often these foods were consumed (daily, weekly, every two weeks, monthly or never). Your information was very well used in the nutritional guidance stage, directed to all the women who participated in the study. This guidance aimed to promote the necessary dietary changes, indicating the inclusion, greater frequency of consumption or even the exclusion of foods, in the nutritional guidance stage of our study.

Results:

The representativeness of the studied group should be verified here – characteristics of the studied group while compared with the general population

Reply: The authors wrote in sample size how was selected the candidates of the study to reach a representative sample. And inserted more results about the studied group.

“The sample size was calculated based on a national study that assessed the Brazilian prevalence of micronutrient inadequacy (POF 2017-2018) [17]. Based on these findings, the prevalence of inadequacy for vitamin A, in female population, is 80.1%.

To obtain the sample size with a 95% confidence interval, considering the prevalence of adequacy of 20%, with a sampling error of 5%, 148 women with the recommended daily food intake according to the Institute of Medicine [18] would be needed to conduct the present study.”

The dietary intake of vitamin A should be presented as a characteristic of the studied group and compared between subgroups.

Reply: To meet the reviewer's request, the authors inserted results from assessment of vitamin A dietary of 200 women studied (mean and deviation of the studied group and in each subgroup [NW, OW, OI and OII]) in table 1.

Discussion:

The representativeness of the studied group should be discussed here

Reply: The authors already wrote the representativeness of the studied group in the last revision version; therefore, we inserted more information to contextualize the theme.

Author Contributions:

Authors should properly define the contributions – e.g. what do Authors mean by “validation” if they did not conduct any validation in their study? There are similar doubts in case of “resources” and “data curation”. Only actions associated with conducting this specific study and preparing manuscript should be indicated here.

It seems that contribution of AM and CB was only minor and they did not participate in preparing manuscript. There is a serious risk of a guest authorship procedure which is forbidden. In such case (if they did not participate in manuscript preparation in any way) they should be rather presented in Acknowledgements Section and not be indicated as authors of the study.

Reply: Dear reviewer, the authors decided according to your consideration to remove A.M. as author of the manuscript and indicate her in Acknowledgements Section. However, the author C.B. had an important and crucial role in the study, once she was direct responsible for all attendance (1080 women), application of exclusion criteria (vitamin A food intake), nutritional guidance about vitamin A and collaboration in the draft of the manuscript. So, the authors need to maintain C.B. in the authorship. We expect your understanding about this subject.

Thank you for all consideration and time to review our manuscript.

Best regards,

Reviewer 2 Report

The authors have responded extensively to my comments, and the paper is ready for publication. 

Author Response

Dear reviewer,

Thank you very much for all considerations and revision in our manuscript.

Best regards,

Round 3

Reviewer 1 Report

The manuscript entitled „ Vitamin A deficiency and its association with visceral adiposity in women” presents interesting issue, but some problems should be corrected.

Major:

There is a serious problem with a methodology, as Authors do not want to present how did they obtain average intake from 3 various methods used (as described below).

Introduction:

Authors should include missing reference – ‘In this scenario, women are gaining more and more prominence for having the highest rates of overweight and obesity and in all age groups, when compared to men (ref)’.

Materials and Methods:

It is still hard to guess how did Authors assess diet vitamin A dietary intake, as they declared as “a food intake survey was performed comprising a 24-hour recall using three food records, two records during the week and one record at the weekend, and a food frequency questionnaire”, so they should clearly declare how did they combine vitamin A data from dietary record and food frequency questionnaire. They indicated that they calculated average intake, but we still do not know how – if it was mean of 3 vitamin A values, it is improper approach, so it should be clearly indicated. It is crucial for the studied issue.

Results:

The representativeness of the studied group should be verified here – characteristics of the studied group while compared with the general population

The dietary intake of vitamin A should be presented as a characteristics of the studied group and compared between subgroups

Discussion:

The representativeness of the studied group should be discussed here

Author Response

LETTER TO REVIEWERS

Biomedicines – Manuscript ID 2199431

Title: “Vitamin A deficiency and its association with visceral adiposity in women”

Reviewer 1

The manuscript entitled „ Vitamin A deficiency and its association with visceral adiposity in women” presents interesting issue, but some problems should be corrected.

Major:

There is a serious problem with a methodology, as Authors do not want to present how did they obtain average intake from 3 various methods used (as described below).

Reply: Dear reviewer, to make it well explained and clear how the assessment of dietary vitamin A was carried out, we have written it down in detail below and added some information in the text of the manuscript.

The assessment of dietary vitamin A was performed using the three-day food record (two during the week and one on the weekend) that quantified the average dietary vitamin A. Additionally, we used the 24-hour recall to train the three-day food record, improving the quality of the information, and the food consumption frequency questionnaire, which had the main function of supporting dietary guidance for all the women in the study.

The 24-hour recall was used to obtain information on vitamin A consumption on the day before the interview, considering preparation, information on weight and portion sizes, in grams, milliliters and household measurements. During data collection, each woman was instructed to ensure that no meals or snacks were missed. The main purpose of its application was to enable the study population to correctly complete the instrument that quantified the average intake of vitamin A, which was the three-day food record, with a view to minimizing filling errors and, consequently, improving the quality of the information.

The three-day food record, two days a week and one on the weekend, was the method used to quantify the average intake of vitamin A in the study, as it was carried out at the time of food consumption. Thus, it was not based on the individual's memory, but on actual consumption. This record was completed by the patient, who was trained to write down all the food and drinks consumed over the three days evaluated, and was also instructed to write down the food consumed outside the home. The individual recorded the size of the portion consumed in detail, the name of the preparation, the ingredients in the composition, the brand of the food and the way of preparation. The patients noted details such as the addition of salt, sugar, oil and sauces, whether the peel of the food was ingested and also whether the food or drink consumed was regular, diet or light. To improve the estimation of portion size, the patient had the help of traditionally used household measures and also photo albums of different portion sizes and food models, which were made available by email by the nutritionist responsible for the care. It should be noted that the food record has been the preferred method of many professionals, in which there is a record of the size of the food portion at the time of consumption, being an important characteristic of the method in question, since memory bias is largely minimized.

At the time of consultation, the responses obtained through the three-day food record were entered and plotted in Dietbox software, which calculated the average daily intake of vitamin A, according to the content of vitamin A in foods as published in the Brazilian Table of Food Composition, which integrates the Dietbox software. Serving size was assessed using the Photo Atlas of Food Portion Sizes. Vitamin A intake was compared with the daily intake values recommended by the Institute of Medicine. The cut-off point adopted for the recommended dietary intake of VA was 700 μg/day of retinol equivalent.

The food consumption frequency questionnaire, based on a list of food sources of vitamin A recommended by the International Vitamin A Consultative Group (IVACG), was used to find out how often these foods were consumed (daily, weekly, every two weeks , monthly or never). Your information was very well used in the nutritional orientation stage, directed to all the women who participated in the study. This guidance aimed to promote the necessary dietary changes, indicating the inclusion, greater frequency of consumption or even the exclusion of foods, in the nutritional guidance stage of our study.

Referências bibliográficas:

Institute of Medicine (IOM). Vitamin A. In: Dietary Reference Intakes for Vitamin A, Vitamin K, Arsenic, Boron, Chromium, Copper, Iodine, Iron, Manganese, Molybdenum, Nickel, Silicon, Vanadium, and Zinc. Washington: National Academic Press; 2001. p. 82-161.

Tabela Brasileira de Composição de Alimentos (TBCA). Universidade de São Paulo (USP). Food Research Center (FoRC). Versão 7.2. São Paulo, 2022. [Accessed: 06/11/2022]. Available from: http://www.fcf.usp.br/tbca.

Pinheiro ABV, Lacerda EM, Benzecry EH, et al. Tabela para avaliação de consumo alimentar em medidas caseiras. Vol. 5; 2001. p. 352.

International Vitamin A Consultative Group (IVACG). The Annecy Accords to assess and control vitamin A deficiency: summary of recommendations and clarifications. Washington, DC: IVACG; 2003

Introduction:

Authors should include missing reference – ‘In this scenario, women are gaining more and more prominence for having the highest rates of overweight and obesity and in all age groups, when compared to men (ref)’.

Reply: The authors inserted the reference [3] in the manuscript.

Materials and Methods:

It is still hard to guess how did Authors assess diet vitamin A dietary intake, as they declared as “a food intake survey was performed comprising a 24-hour recall using three food records, two records during the week and one record at the weekend, and a food frequency questionnaire”, so they should clearly declare how did they combine vitamin A data from dietary record and food frequency questionnaire. They indicated that they calculated average intake, but we still do not know how – if it was mean of 3 vitamin A values, it is improper approach, so it should be clearly indicated. It is crucial for the studied issue.

Reply: Dear reviewer, the authors informed in preview reply all details about the methodology to reach the average of vitamin A dietary and how methods were used in this study. And, into the manuscript the authors inserted information to become clearer this subject.

Results:

The representativeness of the studied group should be verified here – characteristics of the studied group while compared with the general population

The dietary intake of vitamin A should be presented as a characteristics of the studied group and compared between subgroups.

Reply: Dear reviewer, the authors inserted in the last revised version the dietary intake of vitamin A (mean and deviation) of the studied group and compared between subgroups (table 1).

“The sample comprised 200 adult women who met the recommended dietary intake of VA (772.3 ± 59.9 μg/day – retinol equivalent).”

Variables

NW

(n = 80)

OW

(n = 40)

OI

(n = 68)

OII

(n = 12)

p-value

Retinol equivalent (μg/day)

795.2 ± 49.9

781.5 ± 34.5

745.8 ± 72.8

740.1 ± 32.3

0.243

Related to 849 women, that were excluded of the study, had only the assessment of vitamin A intake and not the other variables included in the study. However, to your knowledge, dietary vitamin A of these 849 women was 564.3 ± 23.5 µg/day –retinol equivalent. The authors considered it opportune not to include these data.

Discussion:

The representativeness of the studied group should be discussed here.

Reply: Dear reviewer, we included more information in discussion, and

for a better understanding of the representativeness of the sample, we wrote related information in the letter.

As informed in a previous answer, the sample initially selected corresponded to 1080 women, assisted at the Centro Municipal de Saúde Marcolino Candau (CMSMC) in the county of Rio de Janeiro. This health unit serves around 15.000 people a year free of charge, 10.000 of which through the Family Health strategy and more than 5.000 women referring to the Women's Health Program, from which we extracted the sample of our study. The women assisted come from various regions of the county of Rio de Janeiro (RJ), in the city of RJ, southeastern Brazil.

The CMSMC represents the municipality of RJ by approximately 21.5%, in terms of assistance to women, when compared with assistance to women in the entire municipality of RJ, which is composed of 198 health units, which offer comprehensive health care of women in all life cycles.

The choice of the aforementioned health unit for carrying out the present study was due to the fact that the profile of the patients treated at this health unit did not differ from the verified profile for the set of women attended by the health sector in the county of RJ, according to information made available by the Brazilian Ministry of Health.

Referências bibliográficas:

Ministério da Saúde do Brasil, 2021. Accessed 28/02/2023. Saúde RJ - Subsecretaria Geral - Planejamento - Informação SUS - Dados SUS - Assistência Ambulatorial (saude.rj.gov.br)

Centro Municipal de Saúde Marcolino Candau (CMSMC). Accessed 28/02/2023. Sms Cms Marcolino Candau Ap 10 - Rio de Janeiro - RJ - Centro de Saúde, Unidade Básica (ubs.med.br)

Thank you for all consideration and time to review our manuscript.

Best regards,

Round 4

Reviewer 1 Report

I have nothing more to tell to Authors.

Author Response

Dear reviewer,

Thank you for all considerations.

Best regards,

First author